# Comparison of Fracture Load of the Four Translucent Zirconia Crowns

**DOI:** 10.3390/molecules26175308

**Published:** 2021-09-01

**Authors:** Suchada Kongkiatkamon, Kittipong Booranasophone, Apichat Tongtaksin, Valailuck Kiatthanakorn, Dinesh Rokaya

**Affiliations:** 1BDMS Wellness Clinic, Bangkok Dusit Medical Services PCL, Bangkok 10330, Thailand; kittipong_b@yahoo.com (K.B.); cocriko.at@gmail.com (A.T.); valailuck.kia@bdmswellness.com (V.K.); 2Department of Clinical Dentistry, Walailak University International College of Dentistry, Walailak University, Bangkok 10400, Thailand; dinesh.ro@wu.ac.th

**Keywords:** zirconia, CAD/CAM, crown, dental ceramics, fracture load

## Abstract

Recently, translucent zirconia has become the most prevalent material used as a restorative material. This study aimed to compare the crown fracture load of the four most common different translucent zirconia brands available in the market at 1.5 mm thickness. Standardized tooth preparations for a full ceramic crown were designed digitally with software (AutoCAD) by placing a 1.0 mm chamfer margin and 1.5 mm occluso-cervical curvature for the crown sample manufacturing. Stylized crowns were chosen to control the thickness of the crown. The axial and occlusal thickness were standardized to 1.5 mm thickness except at the central pit, which was 1.3 mm thick. The STL file for the tooth dies was prepared using software (3Shape TRIOS^®^ Patient Monitoring, Copenhagen, Denmark). The tooth dies were printed with a resin material (NextDent Model 2.0, Vertex-Dental B.V., Soesterberg, The Netherlands) using a 3D printing software (3D Sprint^®^ Client Version 3.0.0.2494) from a 3D printer (NextDent™ 5100, Vertex-Dental B.V., Soesterberg, The Netherlands). The printing layer thickness was 50 µm. Then, a total of twenty-eight (*N* = 28) stylized crowns were milled out of AmannGirrbach (Amann Girrbach GmbH, Pforzheim, Germany) (*n* = 7), Cercon HT (Dentsply Sirona, Bensheim, Germany) (*n* = 7), Cercon XT (Dentsply Sirona, Bensheim, Germany) (*n* = 7), and Vita YZ XT (Zahnfabrik, Bäd Sackingen, Germany) (*n* = 7). Following sintering the crowns, sandblasting was performed and they were bonded to the tooth dies with the resin cement (RelyX U-200, 3M ESPE, Seefeld, Germany) and permitted to self-cure under finger pressure for 6 min. The crowns were loaded on the occlusal surface in a universal testing machine (MTS Centurion) with a stainless-steel ball indenter (7 mm radius) with a loading rate of 1 mm/min to contact the stylized crowns on each of the four cusps until failure. A rubber sheet (1.5 mm thickness) was positioned between the crown and indenter, which helped with the load distribution. Statistical analysis was done using SPSS version 20 (IBM Company, Chicago, USA). The fracture loads were analyzed using Dunnett’s T3 test, and the number of cracks was analyzed using the Mann–Whitney U test among the groups. The significant level was set at *p* value = 0.05. The mean fracture loads were 3086.54 ± 441.74 N, 4804.94 ± 70.12 N, 3317.76 ± 199.80 N, and 2921.87 ± 349.67 N for AmannGirrbac, Cercon HT, Cercon XT, and Vita YZ XT, respectively. The mean fracture loads for the surfaces with the greatest number of cracks (excluding the occlusal surfaces) were on the lingual surface for AmannGirrbach and Cercon HT, on the distal and mesial for Cercon XT, and on the buccal for Vita YZ XT. We found that the AmannGirrbach had the most overall cracks. Cercon XT had the greatest number of occlusal cracks and appeared to be the most shattered. Cercon HT had the least number of cracks. In conclusion, Cercon HT presented the best strength properties, the highest fracture load, and no visible cracks. AmannGirrbach presented the lowest strength properties.

## 1. Introduction

In recent decades, there has been considerable development in the materials and technologies in prosthetic and restorative dentistry [1,2,3,4]. Esthetic dental restorations with biological properties present a life-like appearance [5]. The all-ceramic/ metal-free zirconia restorations present the best esthetic outcome as they have a similar natural color and light translucency.

The CAD/CAM technique has superior results with more time efficiency, an improvement in cost/effectiveness, and higher-precision prostheses compared to conventional manufacturing techniques [6]. With this technology, the restorations can be fabricated by additive, layer-by-layer, and subtractive manufacturing processes [7,8].

At present, there are various types of zirconia restorations based on strength and opacity due to advances in zirconia manufacturing. Both the opaque zirconia crown and the translucent lithium disilicate crown require thickness (minimum 1.5 mm) for sufficient strength. One important factor for the longevity of zirconia restorations is the mechanical and fracture load properties of the restoration apart from the marginal fit, fracture resistance, and esthetics [9,10,11]. Studies have been performed to study the mechanical properties and clinical performance of dentine bonded crowns [12,13]. 

The proper manipulation of restorations with adequate thickness presents increased fracture strength when bonded to tooth tissue [14,15]. Axial and occlusal reductions of 1.5 mm and 2 mm, respectively, are needed to provide sufficient strength for the restoration and veneering for ceramic restoration and ≥ 1.5 mm for monolithic lithium disilicate crowns [16,17,18]. The zirconia restorations present greater flexural strength (>1000 MPa) compared to the lithium disilicate restorations (~400 MPa) [19,20,21,22]. One of the major advantages of using monolithic zirconia crowns is to avoid the chipping and delamination that occurs when using bi-layer designs [23].

Currently, zirconia can be classified into three groups based on the yttria content as follows [24]: (a) 3 mole % Y-TZP (3Y-TZP; strong and mainly tetragonal) (AmannGirrbach (Amann Girrbach GmbH, Pforzheim; IPS e.max^®^ ZirCad LT and MO, Ivoclar Vivadent; Lava^™^ Plus, 3M; BruxZir^®^, Glidewell Laboratories; and KATANA^™^ HT, Kuraray Noritake), (b) 4 mole % Y-TZP (4Y-TZP; more translucent) (Zpex^®^ 4, Kraun; IPS e.max ZirCAD MT; and KATANA^™^ ST/STML), and (c) 5 mole % Y-TZP (5Y-TZP; most translucent) (e.g., Lava Esthetic; Cercon^®^ XT, 3M; BruxZir Anterior; KATANA™ UT/UTML; and Zpex Smile).

This study aims to compare the crown fracture load of the four most common different translucent zirconia brands (AmannGirrbach, Cercon HT, Cercon XT, and Vita YZ XT) available in the market with 1.5 mm thickness. The hypothesis of this study was there is no difference in the fracture load of the four most common different translucent zirconia brands (AmannGirrbach, Cercon HT, Cercon XT, and Vita YZ XT).

## 2. Materials and Methods

### 2.1. Tooth Preparation and Model Preparation

Tooth preparations for the full ceramic crown were done digitally with software (AutoCAD) by placing a 1.0 mm chamfer margin and 1.5 mm occluso-cervical curvature for the crown sample manufacturing without periodontal ligaments and roots, since the rigidity of the specimen is taken into account [25]. Stylized (a nonrealistic, distinctive design) crowns were chosen to control the thickness of the crown. The axial and occlusal thickness were standardized to 1.5 mm thickness except at the central pit, which was 1.3 mm thickness as shown in Figure 1.

The STL file for the tooth dies was prepared using computer software (3Shape TRIOS^®^ Patient Monitoring, Copenhagen, Denmark). Tooth dies were printed with a resin material (NextDent Model 2.0, Vertex-Dental B.V., Soesterberg, The Netherlands) using a 3D printing software (3D Sprint^®^ Client Version 3.0.0.2494) from a 3D printer (NextDent™ 5100, Vertex-Dental B.V., Soesterberg, The Netherlands). The printing layer thickness was 50 µm. Then, a total of twenty-eight (*N* = 28) stylized crowns were milled out of AmannGirrbach (Amann Girrbach GmbH, Pforzheim, Germany) (*n* = 7), Cercon HT (Dentsply Sirona, Bensheim, Germany) (*n* = 7), Cercon XT (Dentsply Sirona, Bensheim, Germany) (*n* = 7), and Vita YZ XT (Zahnfabrik, Bäd Sackingen, Germany) (*n* = 7). The compositions of each material used in this study with their translucency values are shown in Table 1.

The crowns were sandblasted before being cemented in order to increase the bond strength. Following sintering the crowns, they were bonded to the dies of teeth with resin cement (RelyX U-200, 3M ESPE, Seefeld, Germany) and allowed to self-cure under finger pressure for 6 min. The crowns were loaded on the occlusal surface in a universal testing machine (MTS Centurion) with a stainless-steel ball indenter (7 mm radius) with a loading rate of 1 mm/min to contact the stylized crowns on each of the four cusps until failure [18,29,30]. A rubber sheet (1.5 mm thickness) was positioned between the crown and indenter, which helped with the load distribution by avoiding contact damage.

### 2.2. Statistical Analysis

The descriptive statistics were calculated from the SPSS version 20 (IBM Company, Chicago, USA). The fracture loads were analyzed using Dunnett’s T3 test, and the number of cracks was analyzed using the Mann–Whitney U test among the groups. The significant level was set at *p* value = 0.05. The sample size was calculated using G*Power 3.1.9.7, which obtained a minimum of three per group, and the power was 99% [31]. 

## 3. Results

The fracture load results are shown in Table 2, Figure 2 and Figure 3. The highest fracture load was shown by the Cercon HT. The order of fracture load was Cercon HT, Cercon XT, AmannGirrbach, and Vita ZY XT. Table 3 shows multiple comparisons of the strength of zirconia crowns among various groups. The Cercon HT presented a significantly higher fracture load compared to AmannGirrbach, Cercon XT, and Vita ZY XT (*p* value < 0.0001).

The cracks on the surfaces of zirconia crowns are shown in Figure 4. AmannGirrbach appeared to have the most overall cracks. Cercon XT had the greatest number of occlusal cracks and appeared to be the most shattered. Cercon HT had the least number of cracks. Samples 4–7 of Cercon HT were completely intact with no visible cracks. The surface with the greatest number of cracks (excluding occlusal surfaces) was on lingual for AmannGirrbach and Cercon HT, on distal and mesial for Cercon X, and on buccal for Vita YZ XT. There was a statistically significant difference between AmannGirrbach vs. Cercon HT (*p* value < 0.0001) and Cercon HT vs. Cercon XT (*p* value < 0.0001), while there was no statistical significant difference between Cercon HT vs. Cercon XT (*p* value = 0.844) and Cercon HT vs. Cercon XT (*p* value = 0.0186).

## 4. Discussion

The advancement in zirconia manufacturing technology has developed various new zirconia compositions with esthetic enhancements. Zirconia is polymorphic and can be present in three crystallographic structures with the same chemical composition: monoclinic, tetragonal, and cubic [32,33]. At room temperature, zirconia is present in its monoclinic form and is stable up to 1170 °C [34,35]. Above this temperature, a transformation occurs to the tetragonal phase that is stable up to 2370 °C. Above the temperature 2670 °C, zirconia assumes its cubic form (Figure 5) [32]. On cooling from high temperatures, a cubic–tetragonal transformation takes place with a slight expansion of the unit cell volume.

The aging process results from the progressive spontaneous transformation of the metastable tetragonal phase to the monoclinic phase (t–m transformation) in the presence of water or water vapor at relatively low temperatures (approximately 30 °C up to 400 °C), which is a phenomenon known as hydrothermal or low-temperature degradation. Regarding the aging and degradation of zirconia, some theories have described the aging mechanism. Yttria, which is used as a dopant in zirconia to achieve phase stability, is exhausted through a reaction with water in the presence of moisture, resulting in a tetragonal to monoclinic (t–m) phase transformation [36,37,38]. 

The zirconia bond of Zr–O is disturbed by water, thereby, creating stress due to the diffusion of OH^−^ and forming lattice defects that cause the t–m phase transformation [39]. O_2_ derived from water likely fills the oxygen vacancies of the matrix and induces a t–m phase transformation [40]. This transformation is associated with micro-cracks and grain pull out that are initially produced on the surfaces of zirconia. Water may penetrate these cracks and promote surface degradation to expand further into the whole material [36]. Sato and Shimada [41] proposed stress corrosion by water, e.g., water reacts with Zr-O-Zr bonds at the crack tips, and Zr-OH bonds are formed according to
-Zr-O-Zr- + H2O → -Zr-OH + HO-Zr-(1)

This study aimed to compare the crown fracture load of the four most common different translucent zirconia brands available on the market; AmannGirrbach (Amann Girrbach GmbH, Pforzheim, Germany), Cercon HT (Dentsply Sirona, Bensheim, Germany), Cercon XT (Dentsply Sirona, Bensheim, Germany), and Vita YZ XT (Zahnfabrik, Bäd Sackingen, Germany). Based on the results, the hypothesis that there is no difference in the fracture load of the four most common different translucent zirconia brands was approved. We found that the Cercon HT presented the best strength properties, the highest fracture load, and no visible cracks. AmannGirrbach presented the lowest strength properties. The reason that Cercon HT showed more fracture load/resistance may be due to the lower yttria content compared with the other groups. Cercon XT and Vita YZ XT had higher yttria contents and had less phase transformations and less fracture resistance.

The yttria contents of AmannGirrbach and Cercon HT are similar; however, the fracture toughness of AmannGirrbach was lower than Cercon HT. The reason could be from the process of the manufacturing process for making translucent zirconia, such as by adding more cubic phases to increase the translucency without changing the phase transformation. Similarly, although the yttria contents of various crown materials are close to each other with similar load applications, the Cercon HT could receive more loads, which could be due to the lowe yttria content being converted to the monoclinic phase.

There are several methods for making translucent zirconia, such as by increasing the stabilized yttrium, increasing the cubic phase ratio without reducing the toughness, and reducing its grain size [24]. Hence, the major difference between the conventional and translucent zirconia materials is the presence of a more stabilized yttrium with increasing the cubic phase ratio. AmannGirrbach’s yttria content is closer to Cercon HT, but it showed a lower load and translucency, possibly due to the composition. Various factors influence the fracture load of a zirconia crown, such as the composition, crown thickness, surface treatments, and cementation technique [29,30,42,43].

Lawson et al. [29] performed a study of the surface treatment and cement selection on various zirconia crowns. They used 3 mol% and 5 mol% yttria partially stabilized zirconia (3Y-PSZ and 5Y-Z) and lithium disilicate crowns and tested their fracture load. All zirconia crowns were luted to resin composite dies with resin-modified glass-ionomer or resin cement. They found that the materials and cement showed significant differences for the fracture load; however, the surface treatments did not. 

Both zirconia crowns luted with resin cement showed a greater fracture load than the resin-modified glass-ionomer cement. In addition, the 3Y-PSZ had a greater fracture load than the 5Y-Z and lithium disilicate for the resin cement. Another study found that the lithium disilicate restorations showed significantly greater load-bearing properties compared wtih 5Y-PSZ but was similar to 4Y-PSZ [43]. In our study, we used only one method to control the cement type, and the fracture load was comparatively higher in their studies. Furthermore, the die resin material used in the study can be compared with the mechanical properties of dentin or enamel mechanical properties. The modulus of elasticity of enamel is 84 GPa and dentine is 18.6 GPa [44], whereas that of epoxy resin is around 35.9 GPa [45]. Even though, the elastic modulus of epoxy resin differs from that of dentin and enamel, it should be still providing a good representation of the test samples.

Similarly, Kim et al. [30] compared the fracture load on the coping thickness and the facial collar design of zirconia crowns where they divided the study into four groups: the standard coping group (thickness of 0.5 mm and facial collar height of 0.2 mm), collarless coping group (thickness of 0.5 mm and no facial collar), modified thicker coping group (thickness of 0.7 mm and facial collar height of 0.2 mm), and thicker coping group (thickness of 0.7 mm including collar height). They found no significant difference in the fracture load between the standard coping and collarless coping groups. They concluded that the thicker coping had the greatest fracture strength compared to the standard and collarless copings.

In our study, the thickness had a 1.0 mm chamfer margin and 1.5 mm incisal-cervical curvature. Hence, the fracture load was higher in our study compared to their study. However, a study by Nakamura et al. [18] mentioned that a monolithic zirconia restoration with 0.5 mm chamfer width and 0.5 mm thickness at the occlusal surface can be used in the posterior teeth for enough fracture resistance. The translucent zirconia can be performed in a small thickness of 1.5 mm [46].

Lopez-Suarez et al. [1] compared the fracture properties (load and pattern) of monolithic and veneering zirconia on posterior restorations fabricated from Lava Zirconia and Lava Plus, respectively. The fracture load values were clinically acceptable. The fracture load showed no difference between the groups. The veneering ceramic was fractured before the fracture of the framework, and various fracture patterns were seen. 

Similarly, Alshiddi et al. [47] studied the effect of size/dimension and the micro-cracks produced from diamond burs during the milling process on the fracture resistance of CAD/CAM fabricated implant-supported cantilever zirconia frameworks. They found significant variations in the fracture load between the implant-supported cantilever zirconia frameworks with different cantilever lengths and thicknesses of the distal abutments were found. Increasing the distal abutment thickness only by 0.5 mm strengthened the distal abutments by significantly increasing the fracture load of the implant-supported cantilever zirconia frameworks.

Some limitations of this study include that we could not create more samples due to financial restrictions. Longer fatigue would be helpful for better prediction of the performance of the crowns, and the differences in the material composition can affect the fracture load results. Further studies can be done to study the effects of the cement on the fracture load of these ceramic crowns. In addition, the fracture load can be studied by comparing the studied zirconia crowns of varying thicknesses.

## 5. Conclusions

The crown fracture loads of the studied four most-common different translucent zirconia brands (AmannGirrbach, Cercon HT, Cercon XT, and Vita YZ XT) were in the acceptable range. Compared to the maximum bite force, all materials presented an acceptable fracture load value. Cercon HT presented the best strength properties, the highest fracture load, and no visible cracks compared to the other three zircona brands. AmannGirrbach presented the lowest strength properties.

## Figures and Tables

**Figure 1 molecules-26-05308-f001:**
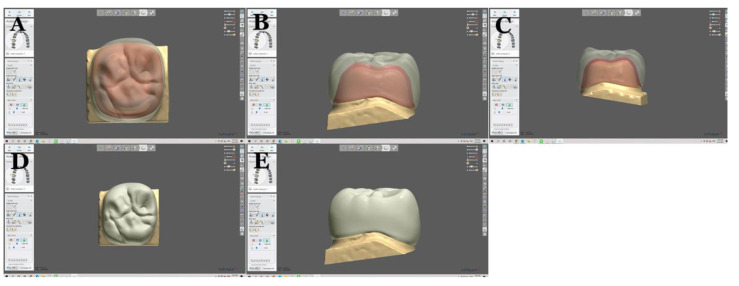
Crown designed digitally using 3Shape software(TRIOS^®^). Crown from different views (**A**–**E**).

**Figure 2 molecules-26-05308-f002:**
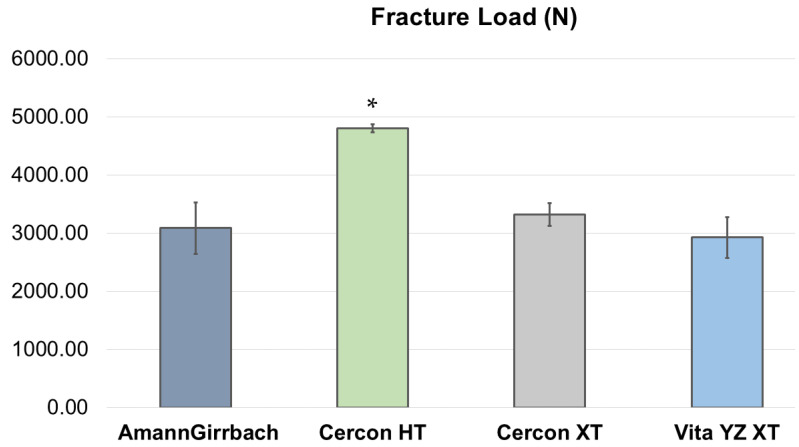
The mean fracture load of various zirconia; AmannGirrbach (Amann Girrbach GmbH, Pforzheim, Germany), Cercon HT (Dentsply Sirona, Bensheim, Germany), Cercon XT (Dentsply Sirona, Bensheim, Germany), and Vita YZ XT (Zahnfabrik, Bäd Sackingen, Germany). * Cercon HT presented a significantly higher fracture load compared to AmannGirrbach, Cercon XT, and Vita ZY XT (*p* value < 0.0001).

**Figure 3 molecules-26-05308-f003:**
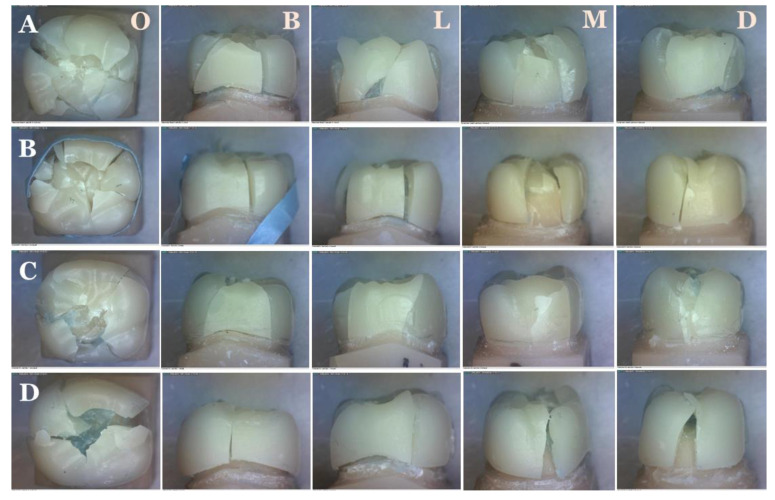
Crown fracture under load. AmannGirrbach (Amann Girrbach GmbH, Pforzheim, Germany) (**A**), Cercon HT (Dentsply Sirona, Bensheim, Germany), (**B**), Cercon XT (Dentsply Sirona, Bensheim, Germany) (**C**), and Vita YZ XT (Zahnfabrik, Bäd Sackingen, Germany) (**D**). O = Occlusal, B = Buccal, L = Lingial, M = Mesial, and D = Distal.

**Figure 4 molecules-26-05308-f004:**
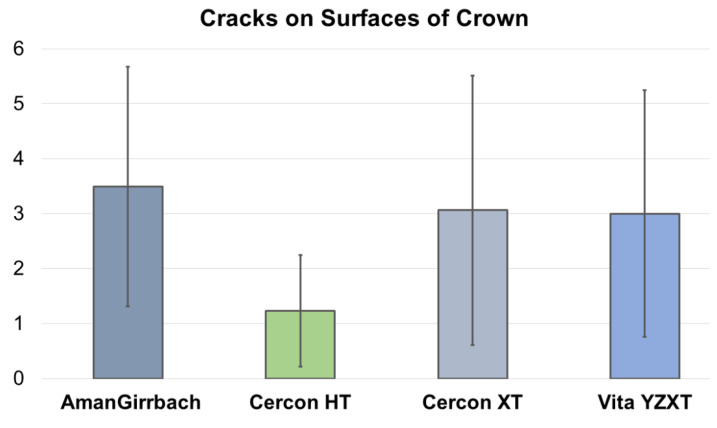
Cracks on surfaces of zirconia crowns.

**Figure 5 molecules-26-05308-f005:**
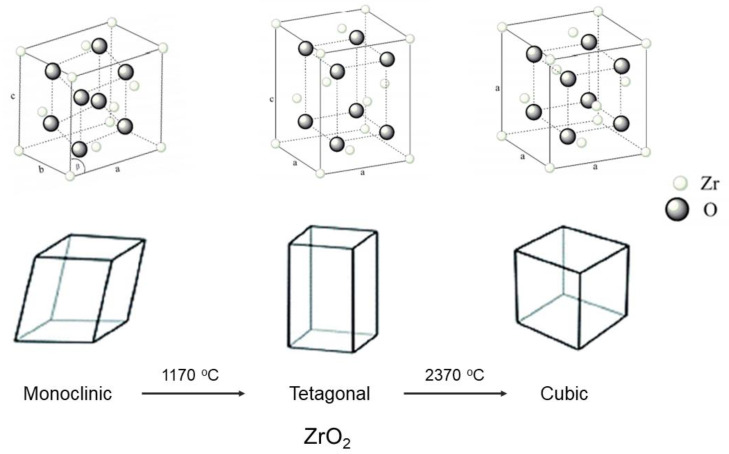
Structure of zirconia (monoclinic, tetragonal, and cubic). Modified from [32].

**Table 1 molecules-26-05308-t001:** Summary of the crown materials used in this study.

Crown Materials (N = 28)	Chemical Composition	Translucency	Manufacturer	Reference
AmannGirrbach (*n* = 7)	ZrO_2_ Y_2_O_3_: 4–5.6% HfO_2_: ≤5% Al_2_O_3_: ≤0.5% Al_2_O_3_, Other oxides including Silicon oxide: <1%	30%	Amann Girrbach GmbH, Pforzheim, Germany	[26]
Cercon HT (*n* = 7)	ZrO_2_ Y_2_O_3_: 5% HfO_2_: <3% Al_2_O_3_, Other oxides including Silicon oxide: <1%	41%	Dentsply Sirona, Bensheim, Germany	[2]
Cercon XT (*n* = 7)	ZrO_2_ Yttrium oxide (Y_2_O_3_): 9% HfO_2_: <3 % Al_2_O_3_, Other oxides including Silicon oxide: <1%	49%	Dentsply Sirona, Bensheim, Germany	[27]
Vita YZ XT (*n* = 7)	ZrO_2_ Yttrium oxide: 8–10% HfO_2_: 1–3% Al_2_O_3_, Other oxides including Silicon oxide: <1%	50%	Zahnfabrik, Bäd Sackingen, Germany	[28]

ZrO_2_ = Zirconium oxide, Y_2_O_3_ = Yttrium oxide, HfO_2_ = Hafnium oxide, Al_2_O_3_ = Aluminium oxide.

**Table 2 molecules-26-05308-t002:** Descriptive statistics of the strength of zirconia crowns (load recorded in Newtons).

Specimen	AmannGirrbach (*n* = 7)	Cercon HT (*n* = 7)	Cercon XT (*n* = 7)	Vita YZ XT (*n* = 7)
1	3116.44	4694.21	3395.24	2647.03
2	2426.12	4922.22	3252.99	2750.66
3	3425.61	4752.21	3376.96	3124.21
4	2526.83	4823.54 ^ψ^	3479.74	3081.29
5	3234.94	4807.57 ^ψ^	3581.93	3309.39
6	3573.10	4821.90 ^ψ^	3122.34	3198.18
7	3302.76	4812.91 ^ψ^	3015.11	2342.32 ^§^
Mean	3086.54	4804.94	3317.76	2921.87
SD	441.74	70.12	199.80	349.67
CV (%)	14.31%	1.46%	6.02%	11.97%

CV = Coefficient of variation, SD = Standard deviation. ^ψ^ Tests stopped at these values and the crowns had cracks but did not beak all the way. ^§^ Crown had a gap at the margin due to cementing error.

**Table 3 molecules-26-05308-t003:** Multiple comparisons of the fracture load of zirconia crowns among various groups.

	AmannGirrbach	Cercon HT	Cercon XT	Vita YZ XT
AmannGirrbach	-	<0.0001 *	0.748	0.961
Cercon HT	<0.0001 *	-	<0.0001 *	<0.0001 *
Cercon XT	0.748	<0.0001 *	-	0.135
Vita YZ XT	0.961	<0.0001 *	0.135	-

Statistical analysis was performed with Dunnett’s T3 test. * Significant differences at *p* value < 0.05.

## Data Availability

The data presented in this study are available on request from the corresponding authors.

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
