# Peer review of "Comparison of Fracture Load of the Four Translucent Zirconia Crowns"

_molecules, 2021, doi:10.3390/molecules26175308_

Round 1

Reviewer 1 Report

Dear. Author,

Please find an attached file.

The article can be accepted for publication after minor revision.

Author Response

Response to Reviewer 1 Comments

Thank you for your positive comments. Corrections in the Manuscript for Reviewer 1 are highlighted in Yellow color.

Page 2. And 3.

Materials and methods,

Why did you not use sand abrasion and acidic monomer on the inner surface of crowns before there were bonded with resin cement?

The results of fracture load are influenced by mechanical retention and

You should mention that reason on M & M or discussion.

Response: The crowns were sand blasted before cemented in order to increase the bond strength. (Added in the Materials and Methods, Page 3)

Page 3.

There were were bonded to the dies.

You should delete “were”.

Response: Deleted the repeated words.

Page 3.

Data was analyzed by one-way ANOVA and Tukey’s HSD test.

However, you should not use ANOVA and Tukey’s HSD because data of fracture load indicate non-homogeneity of variances in Yellow high-light.

I strongly suggest that data should be analyzed by Dunnett’s T3 test only using SPSS.

And replace “one-way ANOVA and Tukey’s HSD test” with Dunnett’s T3 test.

Levene's Test of Equality of Error Variancesa,b

Levene Statistic

df1

df2

Sig.

data

Based on Mean

5.881

3

24

.004

Based on Median

2.247

3

24

.109

Based on Median and with adjusted df

2.247

3

14.034

.128

Based on trimmed mean

5.439

3

24

.005

Tests the null hypothesis that the error variance of the dependent variable is equal across groups.

Response: The data were analyzed by Dunnett’s T3 test using SPSS.

Multiple Comparisons

Load

Dunnett T3

(I) Groups

(J) Groups

Mean Difference (I-J)

Std. Error

Sig.

95% Confidence Interval

Lower Bound

Upper Bound

Amangirrbach

Cercon HT

-1718.394*

169.052

.000

-2327.74

-1109.05

Cercon XT

-231.216

183.246

.748

-841.17

378.74

Vita YZXT

164.674

212.939

.961

-500.06

829.41

Cercon HT

Amangirrbach

1718.394*

169.052

.000

1109.05

2327.74

Cercon XT

1487.179*

80.032

.000

1212.85

1761.51

Vita YZXT

1883.069*

134.793

.000

1401.58

2364.55

Cercon XT

Amangirrbach

231.216

183.246

.748

-378.74

841.17

Cercon HT

-1487.179*

80.032

.000

-1761.51

-1212.85

Vita YZXT

395.890

152.215

.135

-95.82

887.60

Vita YZXT

Amangirrbach

-164.674

212.939

.961

-829.41

500.06

Cercon HT

-1883.069*

134.793

.000

-2364.55

-1401.58

Cercon XT

-395.890

152.215

.135

-887.60

95.82

*. The mean difference is significant at the 0.05 level.

Page 3 and other pages.

Please replace “P value” with “p value”.

Response: P value changed to p value in all pages.

Page 4.

“SD=Mean” indicates in Table 1.

What does it mean?

Response: Corrected. There was a mistake. SD = Standard deviation

Page 4.

“P=<0.0001”

You should use small “p” and remove “=”.

Response: Corrected.

Page 6.

Average and SD of number of cracks are observed in Figure 4.

Distribution of these data indicate non-normal in your study.

I strongly suggest that data should be analyzed by Mann-Whitney U test with Bonferroni correction using SPSS.

Response:

The number of cracks were analyzed by Mann-Whitney U test using SPSS.

Page 7, in Line 8.

Reference No. “29-231” is not correct.

Response: Reference corrected.

References

Ref. No.19 is same Ref. as 35.

There are many grammatical and even more numerous spelling errors in manuscript and references. These errors should be corrected.

Response: Reference corrected. Grammatical and even spelling errors in manuscript and references are removed.

Page 8.

There is mentioned about comparison between fracture load of Zirconia and PEEK crown in discussion.

You should remove this sentence from discussion.

Because it is same thing as comparing between Metal materials and resin materials.

Response: The sentence and the references are removed.

Reviewer 2 Report

The present study aimed to compare the crown fracture load of the four different translucent zirconia brands available in the market with 1.5 mm thickness. The subject is interesting, however the small crack propagation and fatigue effect cannot be evaluated in studies with this methodological approach.  In the oral environment, the restorations will fail by cycling chewing loads and not by the maximum fracture load. Despite these limitations, I believe that the authors can give a proper direction in the text flow when performing the required suggestions:

This is a very simple manuscript; therefore, the study type should be changed from “article” to “communication”.

Title:

The word “types” erroneously give the idea that you are testing different crown designs. Please correct it.

Abstract:

Describe the tooth that was prepared during the samples manufacturing;

Insert sample size information;

Insert the load to fracture in N and standard deviation for each group;

Introduction:

One of the major advantage of using monolithic zirconia crowns is to avoid the chipping and delamination that occurs when using bi-layer designs. Please improve your introduction with this information using the following manuscript:

Lima JC, Tribst JP, Anami LC, et al. Long-term fracture load of all-ceramic crowns: Effects of veneering ceramic thickness, application techniques, and cooling protocol. J Clin Exp Dent. 2020;12(11):e1078-e1085. Published 2020 Nov 1. doi:10.4317/jced.57352

Describe the major composition differences between conventional and translucent zirconia materials;

“The zirconia restorations present greater flexural strength (>1000 MPa) compared to the lithium disilicate restorations (~400 MPa) [15,16,17,18].” Is this information valid for all translucent zirconia materials?

“…zirconia has become the most prevalent material used as a restorative material…” Insert a reference for this statement or remove it.

Insert your study hypothesis.

Methods:

Define “Stylized crowns”;

Describe the prepared tooth;

Describe the die resin mechanical properties and correlate it with dentin or enamel mechanical properties.

Did you have to create roots and simulate periodontal ligament during the samples manufacturing?  Check the following reference and insert this information in the methods:

Dal Piva AO, Tribst JP, Borges AL, de Melo RM, Bottino MA. Influence of substrate design for in vitro mechanical testing. J Clin Exp Dent. 2019 Feb 1;11(2):e119-e125. doi: 10.4317/jced.55353. PMID: 30805115; PMCID: PMC6383903.

“A rubber sheet (1.5 mm thickness) was positioned between the crown and indenter which helped with the load distribution.” Explain how.

Describe the sample size calculation and statistical test power.

Discussion:

Explain if your study hypothesis has been denied or accepted.

The first two paragraphs of your discussion section are just introduction about zirconia material. Please discuss your hypothesis, methods or results with the reported information.

According to the literature, the translucent zirconia can be performed in small thickness than 1.5 mm. Improve your discussion with this information considering the following manuscripts:

Dal Piva AMO, Tribst JPM, Benalcázar Jalkh EB, Anami LC, Bonfante EA, Bottino MA. Minimal tooth preparation for posterior monolithic ceramic crowns: Effect on the mechanical behavior, reliability and translucency. Dent Mater. 2021 Mar;37(3):e140-e150. doi: 10.1016/j.dental.2020.11.001. Epub 2020 Nov 25. PMID: 33246664.

“Some limitations of this study are we could not do more samples due to finical constraints.” What is finical constraints?

The PEEK information is out of context at the end of discussion section, please remove it.

Conclusion

Comparing to the maximum bite force, all materials would present an acceptable fracture load value. Improve your conclusion with this information.

Author Response

Response to Reviewer 2 Comments

Thank you for your positive comments. Corrections in the Manuscript for Reviewer 1 are highlighted in Green color.

The present study aimed to compare the crown fracture load of the four different translucent zirconia brands available in the market with 1.5 mm thickness. The subject is interesting, however the small crack propagation and fatigue effect cannot be evaluated in studies with this methodological approach.  In the oral environment, the restorations will fail by cycling chewing loads and not by the maximum fracture load. Despite these limitations, I believe that the authors can give a proper direction in the text flow when performing the required suggestions:

This is a very simple manuscript; therefore, the study type should be changed from “article” to “communication”.

Response: The study type is changed to Communication.

Title:

The word “types” erroneously give the idea that you are testing different crown designs. Please correct it.

Response: Title is changed to “Comparison of Fracture Load of the Four Translucent Zirconia Crowns”.

Abstract:

Describe the tooth that was prepared during the samples manufacturing;

Insert sample size information;

Insert the load to fracture in N and standard deviation for each group;

Response: Tooth that was prepared during the samples manufacturing Abstract and Page 2).

Sample size distribution is added (Abstract, Page 2, Table 1 and 2).

The load to fracture in N and standard deviation for each group is added in the abstract.

Introduction:

One of the major advantage of using monolithic zirconia crowns is to avoid the chipping and delamination that occurs when using bi-layer designs. Please improve your introduction with this information using the following manuscript:

Lima JC, Tribst JP, Anami LC, et al. Long-term fracture load of all-ceramic crowns: Effects of veneering ceramic thickness, application techniques, and cooling protocol. J Clin Exp Dent. 2020;12(11):e1078-e1085. Published 2020 Nov 1. doi:10.4317/jced.57352

Response: In introduction is improved by adding the information.

Describe the major composition differences between conventional and translucent zirconia materials;

Response: The major differences between the conventional and translucent zirconia materials is the presence of more stabilized yttrium with increasing the cubic phase ratio (Page 7).

“The zirconia restorations present greater flexural strength (>1000 MPa) compared to the lithium disilicate restorations (~400 MPa) [15,16,17,18].” Is this information valid for all translucent zirconia materials?

Response: This information is generally applicable for all translucent zirconia.

“…zirconia has become the most prevalent material used as a restorative material…” Insert a reference for this statement or remove it.

Response: This line is removed.

Insert your study hypothesis.

Response: Hypothesis is added (Page 2 and 8).

Methods:

Define “Stylized crowns”;

Describe the prepared tooth;

Describe the die resin mechanical properties and correlate it with dentin or enamel mechanical properties.

Response: Stylized means nonrealistic, distinctive design crown.

Tooth preparations for full ceramic crown were done digitally with software (AutoCAD) by placing a 1.0 mm chamfer margin and 1.5 mm occluso-cervical curvature for the crown sample manufacturing without periodontal ligaments and roots, since the rigidity of the specimen is taken into account [21]. (Method)

The die resin mechanical properties are compared with the dentin or enamel mechanical properties (Page 8).

Did you have to create roots and simulate periodontal ligament during the samples manufacturing?  Check the following reference and insert this information in the methods:

Dal Piva AO, Tribst JP, Borges AL, de Melo RM, Bottino MA. Influence of substrate design for in vitro mechanical testing. J Clin Exp Dent. 2019 Feb 1;11(2):e119-e125. doi: 10.4317/jced.55353. PMID: 30805115; PMCID: PMC6383903.

Response: The information is added in the method.

“A rubber sheet (1.5 mm thickness) was positioned between the crown and indenter which helped with the load distribution.” Explain how.

Response: A rubber sheet (1.5 mm thickness) was positioned between the crown and indenter which helped with the load distribution by avoiding contact damage (Page 3).

Describe the sample size calculation and statistical test power.

Response: The sample size was calculated using G*Power 3.1.9.7 and obtained a minimum of 3 per each group and the power of the was 99%.

Discussion:

Explain if your study hypothesis has been denied or accepted.

The first two paragraphs of your discussion section are just introduction about zirconia material. Please discuss your hypothesis, methods or results with the reported information.

Response: The hypothesis is added (Page 2) and it is denied (Page 8).

According to the literature, the translucent zirconia can be performed in small thickness than 1.5 mm. Improve your discussion with this information considering the following manuscripts:

Dal Piva AMO, Tribst JPM, Benalcázar Jalkh EB, Anami LC, Bonfante EA, Bottino MA. Minimal tooth preparation for posterior monolithic ceramic crowns: Effect on the mechanical behavior, reliability and translucency. Dent Mater. 2021 Mar;37(3):e140-e150. doi: 10.1016/j.dental.2020.11.001. Epub 2020 Nov 25. PMID: 33246664.

Response: The information is added.

“Some limitations of this study are we could not do more samples due to finical constraints.” What is finical constraints?

Response: The limitation is improved to “financial restriction”.

The PEEK information is out of context at the end of discussion section, please remove it.

Response: The PEEK information is removed.

Conclusion

Comparing to the maximum bite force, all materials would present an acceptable fracture load value. Improve your conclusion with this information.

Response: The conclusion is improved.

Round 2

Reviewer 2 Report

I am satisfied with the corrections and consider the manuscript ready for publication.
Prior to reference 46, in the discussion, the sentence "Improve your discussion with this information by considering the following manuscripts" should be removed.